# "Can't live willingly": A thematic synthesis of qualitative evidence exploring how early marriage and early pregnancy affect experiences of pregnancy in South Asia

**Faith A. Miller** [1]*, **Sophiya Dulal** [2], **Anjana Rai**[3], **Lu Gram**[1], **Helen Harris-Fry** [4], **Naomi M. Saville**[1]

1 Institute for Global Health, University College London, London, United Kingdom, 2 School of Health Sciences, Western Sydney University, Sydney, Australia, 3 School of Public Health and Social Work, Queensland University of Technology, Brisbane, Australia, 4 Department of Population Health, The London School of Hygiene & Tropical Medicine, London, United Kingdom

* faith.miller.19@ucl.ac.uk

**Data Availability Statement:** The data that support the findings of this study were extracted from the

## Abstract

In South Asia, early marriage has been associated with a range of adverse outcomes during pregnancy and infancy. This may partly be explained by early marriage leading to a younger maternal age, however it remains unclear which other factors are involved. This review aimed to synthesise the qualitative evidence on experiences of pregnancy following early marriage or early pregnancy in South Asia, to inform our understanding of the mechanisms between early marriage and adverse pregnancy outcomes. We searched MEDLINE, EMBASE, Scopus, Global Index Medicus, CINAHL, PsycINFO, Web of Science, and grey literature on 29/11/2022 to identify papers on experiences of pregnancy among those who married or became pregnant early in South Asia (PROSPERO registration number: CRD42022304336, funded by an MRC doctoral training grant). Seventy-nine papers from six countries were included after screening. We appraised study quality using an adapted version of the Critical Appraisal Skills Programme tool for qualitative research. Reporting of reflexivity and theoretical underpinnings was poor. We synthesised findings thematically, presenting themes alongside illustrative quotes. We categorised poor pregnancy experiences into: care-seeking challenges, mental health difficulties, and poor nutritional status. We identified eight inter-connected themes: restrictive social hierarchies within households, earning social position, disrupted education, social isolation, increased likelihood of and vulnerability to abuse, shaming of pregnant women, normalisation of risk among younger women, and burdensome workloads. Socioeconomic position and caste/ethnic group also intersected with early marriage to shape experiences during pregnancy. While we found differences between regions, the heterogeneity of the included studies limits our ability to draw conclusions across regions. Pregnancy experiences are largely determined by social hierarchies and the quality of relationships within and outside of the household. These factors limit the potential for individual factors, such as education and empowerment, to improve experiences of pregnancy for girls married early.

'results' section of the published literature referenced in this review. Their accessibility depends on journal open access policies.

**Funding:** This work was supported by the Medical Research Council (MR/N013867/1 to FM). The funders had no role in study design, data collection and analysis, decision to publish, or preparation of the manuscript.

**Competing interests:** The authors have declared that no competing interests exist.

# Introduction

## Early marriage

Early marriage is defined as a marriage or informal union in which one or both partners is below the age of 18 years [1]. While rates of early marriage have decreased globally from an estimated 25% to 20% among girls since 2010, it remains common in South Asia, where around one third of marriages occur during childhood [2]. Early marriage has been associated with adverse maternal and infant health outcomes such as pre-eclampsia, antepartum haemorrhage, delivering a low birthweight baby, preterm delivery, and infant mortality [3–6]. For some outcomes, this may be partly explained by an increased likelihood of becoming pregnant at a young age, however there are also independent effects of early marriage, although the mechanisms are poorly understood [5, 7, 8]. Understanding the mechanisms between early marriage and adverse maternal and infant health is critical to understand what support is needed to improve the health and wellbeing of girls married early [9].

Early marriage is intimately connected with local cultural and gender norms which simultaneously drive the practice of early marriage and shape experiences [10, 11]. In alignment with these norms, households tend to have hierarchies relating to gender and age, meaning young married girls are often discriminated against [12, 13]. In South Asia, early marriage has been associated with interrupted education, restricted mobility outside of the home, social isolation, and limited decision-making [14–17]. This can negatively impact care-seeking during pregnancy [18]. Girls married early are also more vulnerable to gender-based violence within their marital homes [19]. Young brides in South Asia report lower uptake of contraception and higher rates of pregnancy termination than those marrying in adulthood, highlighting the adverse reproductive health sequelae [20, 21]. Inter-connected with each of these experiences is mental health; while large studies are lacking, there is increasing evidence that early marriage has negative mental health consequences [22, 23]. This highlights the complexity which must be considered when exploring outcomes from early marriage.

Over recent decades, the evidence exploring consequences of early marriage has grown. Quantitative reviews support an association between early marriage and a range of adverse reproductive and maternal health outcomes, but highlight the dearth of evidence exploring the mechanisms involved [24–26]. Qualitative studies are well suited to explore such mechanisms, as they centre upon the lived experiences, motivations, and behaviours of those affected [27]. South Asia is a culturally diverse region, represented by its linguistic, religious, ethnic, and geographical diversity between and within countries [28, 29]. Furthermore, differing rates of socioeconomic development and demographic transitions in different countries and subnational regions mean there are differences in the health and social challenges faced across South Asia [30]. However, similar social structures, norms, and values persist across South Asia which affect experiences of early marriage [23–25, 31]. Therefore, summaries of context-specific evidence are required. Seeking to address this gap, this review aims to synthesise the qualitative literature from South Asia on how early marriage and early pregnancy interact in shaping pregnancy experiences. We aim to provide insight on the mechanisms through which early marriage may affect maternal health outcomes.

# Methods

The review protocol was registered on PROSPERO on 10/02/2022 (registration number: CRD42022304336) [32].

**Table 1. PICOS framework to determine eligibility.**

| | |
|---|---|
| Participants | Study conducted in at least one South Asian country (According to the UN classification: Afghanistan, Bangladesh, Bhutan, India, Maldives, Nepal, Pakistan, and Sri Lanka [33]) Participants with first- or second-hand experience of pregnancy or childbirth (current or previous); may include partners, parents, parents-in-law, healthcare providers |
| Intervention/ Exposure | Marriage or pregnancy at an early age as characterised by the authors (including but not limited to <18 years, 'adolescent', 'early', 'young') |
| Comparison | Marriage or pregnancy at a later age *Reference to a comparator group may be implied* |
| Outcome | Experiences of pregnancy or childbirth, including but not limited to nutrition, psychosocial health, care-seeking, and family relationships |
| Study type | Studies reporting primary data with a qualitative methodology, including mixed-methods studies with a qualitative component *Brief answers to open-ended survey questions are not considered qualitative* |

## Eligibility criteria

We used the PICOS framework to outline eligibility criteria, summarised in Table 1.

## Search strategy

Searches were conducted by FM using Ovid (MEDLINE), EMBASE, Scopus, Global Index Medicus, CINAHL, PsycINFO, Web of Science, and the ProQuest Dissertation & Theses Global database. Searches were run on 31/01/2022 and updated on 29/11/2022, using a combination of free text terms, Medical Subject Heading terms, and database-specific limiters, with no language or date restrictions (S1 Appendix). Forward and backward citation searching was undertaken to capture resources citing or being cited by the included literature [34]. FM searched websites of relevant organisations using the terms 'Marriage' and 'Pregnancy' (S1 Appendix).

## Study selection

Two authors screened 10% of titles and abstracts (5% FM & HHF, 5% FM & NS) and disagreements were resolved by discussion. As authors agreed in >90% of papers, FM proceeded to screen the remaining titles and abstracts independently. 10% of full-texts were then screened by two authors (5% FM & HHF, 5% FM & NS), with disagreements resolved by discussion. As authors agreed in >90% of papers, FM screened the remaining full-texts.

## Quality appraisal

We appraised study quality using the Critical Appraisal Skills Programme (CASP) tool for qualitative research, which consists of 10 questions which help appraise the strengths and limitations of qualitative methodology [35]. We selected the CASP tool for its useability and focus on contextualising study findings, however we included an additional question on theoretical underpinnings because the tool has been criticised for not being sensitive to the theoretical validity of studies [35–37]. FM and LG appraised a subset of studies together (10%; 7 papers) before FM appraised the remaining studies independently. Studies were rated 'High', 'Medium', or 'Low' relevance, according to the quality of reporting and relevance to the review question.

## Thematic synthesis

There are a range of approaches to qualitative synthesis, which differ in the way they identify studies for inclusion, examine similarities and differences between studies, appraise study

quality, and go beyond the primary studies to generate additional understanding [38, 39]. Thematic syntheses are similar to meta-ethnography in the way they generate higher level analytical themes from descriptive themes identified within the primary studies, however they use a more clearly defined approach to searching and quality appraisal [40, 41]. Thematic syntheses are well suited to hypothesis generation and synthesising findings from studies spanning multiple disciplines and paradigms, making it well-suited to the aims of this review [42, 43]. However, this approach has been criticised for decontextualising findings, therefore efforts must be made to consider how differences between contexts may explain contradictions between studies [42].

We extracted data on item characteristics, research aims, methods, and findings and entered these into an Excel spreadsheet. We uploaded papers to NVivo 12 for synthesis of the 'results/findings' section of each paper. In two case study papers author interpretation took place in the 'discussion' and therefore the 'discussion' was also extracted for synthesis.

We synthesised findings thematically as follows: i) coding the results of primary studies line-by-line (according to their context), ii) comparing codes between studies (considering similarities and differences), and iii) developing analytical themes (focusing on the review aims to generate a higher-level interpretation, while considering the context of each study to prevent de-contextualisation) [40]. When synthesising findings we sought out contradictory evidence to challenge the themes we were developing.

AR, FM & SD coded 'High'-relevance studies inductively, each coding independently and developing initial themes together through discussion and comparison. FM then deductively coded 'medium'- and 'low'-relevance studies, updating the themes iteratively in discussion with co-authors. This research was primarily undertaken by FM as part of her PhD. As a non-South Asian woman, FM has worked closely with researchers from South Asia (AR and SD) for this review to contextualise findings.

## Results

Of the 6,195 papers identified through database searching, we included 66 (Fig 1). We identified an additional 13 grey literature sources, resulting in 79 sources in total. According to our judgement, >80% of sources adequately reported on their research aims and justified their qualitative approach, whereas <20% adequately reported on positionality and theoretical underpinnings (Fig 2 and S2 Appendix). We deemed 14 sources 'high' relevance [44–57], 39 'medium' [58–96], and 26 'low' [97–122]. 24 studies were conducted in India (10 from Southern/Central India, 10 from Northern India, 4 from multiple sites), 19 from Nepal (11 from Hills/Mountains, 7 from the Terai, 1 from multiple sites), 15 from Bangladesh, 14 from Pakistan, 4 from Afghanistan, 2 from Sri Lanka, and 1 from multiple countries (Fig 3). Notably, no high relevance studies were undertaken in India, whereas a high proportion were undertaken in Bangladesh (n = 6) and Nepal (n = 5). Research sites were in rural settings in 39 studies, urban in 20, and either a combination of both or unspecified in 20. Broadly, research aims focused on care-seeking (n = 27), experience of life during pregnancy (n = 18), sociocultural norms relating to maternal health (n = 17), abuse (n = 8), nutrition (n = 7), and knowledge (n = 2), however studies often reflected on a range of factors (Fig 3). The evidence base has increased with each decade, with two thirds of studies collecting data since 2010. Table 2 summarises the characteristics of high relevance studies (medium and low relevance studies presented in S3 Appendix).

### Themes

We grouped experiences of pregnancy into three categories: care-seeking, mental health, and nutrition (Fig 4). We identified eight themes through which early marriage and early

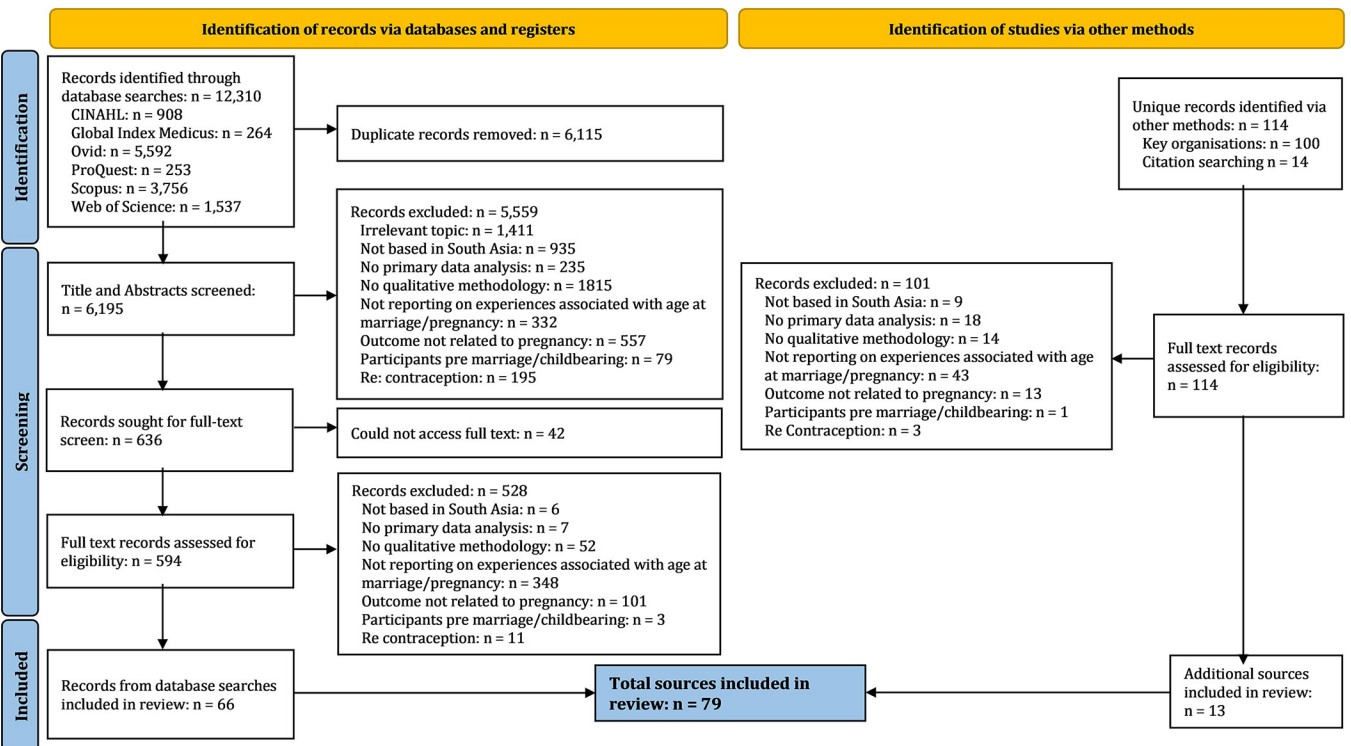

**Fig 1. PRISMA flow diagram presenting screening for the review [123].**

pregnancy affect experiences, relating to: restrictive household hierarchies, earning social position, knowledge of pregnancy needs, social isolation, increased likelihood of and vulnerability to abuse, shaming of pregnant women, normalisation of risk among younger women, and burdensome workloads (S4 Appendix). We narratively describe each theme, the evidence on how early marriage and/or early pregnancy affects each theme, and how the theme affects each category.

**Early marriage intersecting with other identities.** Within each of the following themes, persistent sociocultural hierarchies, such as socioeconomic status and caste group, shaped the pregnancy experiences of women and girls, moderating outcomes following early marriage and early pregnancy.

Women and girls from more disadvantaged caste or ethnic groups faced persistent challenges relating to their health and well-being. They were more likely to experience discrimination from healthcare providers, moderating experiences of abuse following early marriage [44, 46, 63, 71, 107, 113, 122]. However, Mary Cameron found that women and girls from more disadvantaged caste groups in rural Nepal were more involved in choosing a spouse than those from advantaged caste groups, improving marital relationships which were otherwise negatively affected by early marriage [44]. Women from advantaged caste groups also faced tighter restrictions to their movement and behaviour during pregnancy, compounding with the effect of early marriage [44, 91].

Poverty was commonly cited as a driver of early marriage and mediator of experiences. Financial stressors, such as unpaid dowries and debt from loans, increased anxiety during pregnancy for those married early [46, 49, 52, 53, 56, 67, 71, 74, 83, 84, 87, 90, 113]. Many feared they would not be able to seek care due to financial constraints [44, 46, 49, 52, 53, 56, 67, 71, 84, 91], as expressed by this pregnant adolescent:

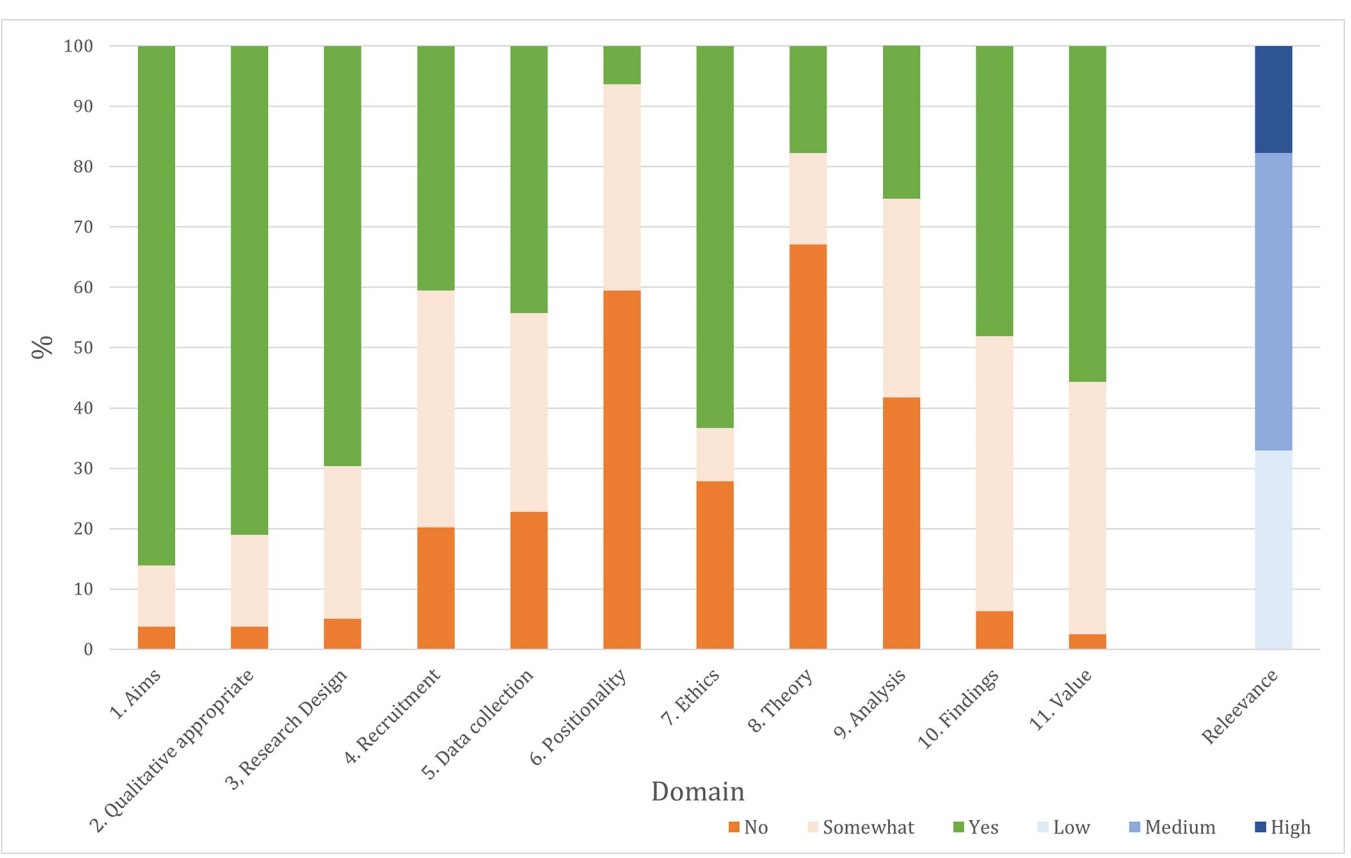

**Fig 2. Summary of responses for each question from the Critical Appraisal Skills Programme (CASP) tool and overall relevance for all included studies (n = 79).**

*"I wanted to call the doctor. I was so sad that my husband said we should wait longer. I was trying so hard. I didn't want to go through so much pain just so we wouldn't have to spend money."* Rural Bangladesh, 2009 [84]

Financial vulnerabilities made pregnant women and girls feel more dependent on their family, exacerbating restrictive hierarchies among those married early [44, 45, 52, 53, 63, 65]. Financial vulnerabilities also limited the options available to women and girls wanting to escape an abusive marriage, which was more common among those married early [52, 53, 63, 65, 83].

However, families often gave financial reasons for not accessing care, despite services being provided for free or there being financial incentives available [44, 49, 56, 85, 90], as reported by this author reflecting on a girl whose father-in-law refused treatment despite offers of free transport:

*"The father-in-law was responsible for the final decision, and he decided against it, saying that "We do not have the money to cover the expenses", even after the local men had offered to carry her there for free... the baby finally came out, stillborn, (and the mother) died."* Rural Nepal, 1988/89 [44]

**Family hierarchy preventing women and girls from speaking up and being listened to.**
The most common theme we identified in this review related to the ways in which social hierarchies, which persist within households across South Asia, limit the ability of women and

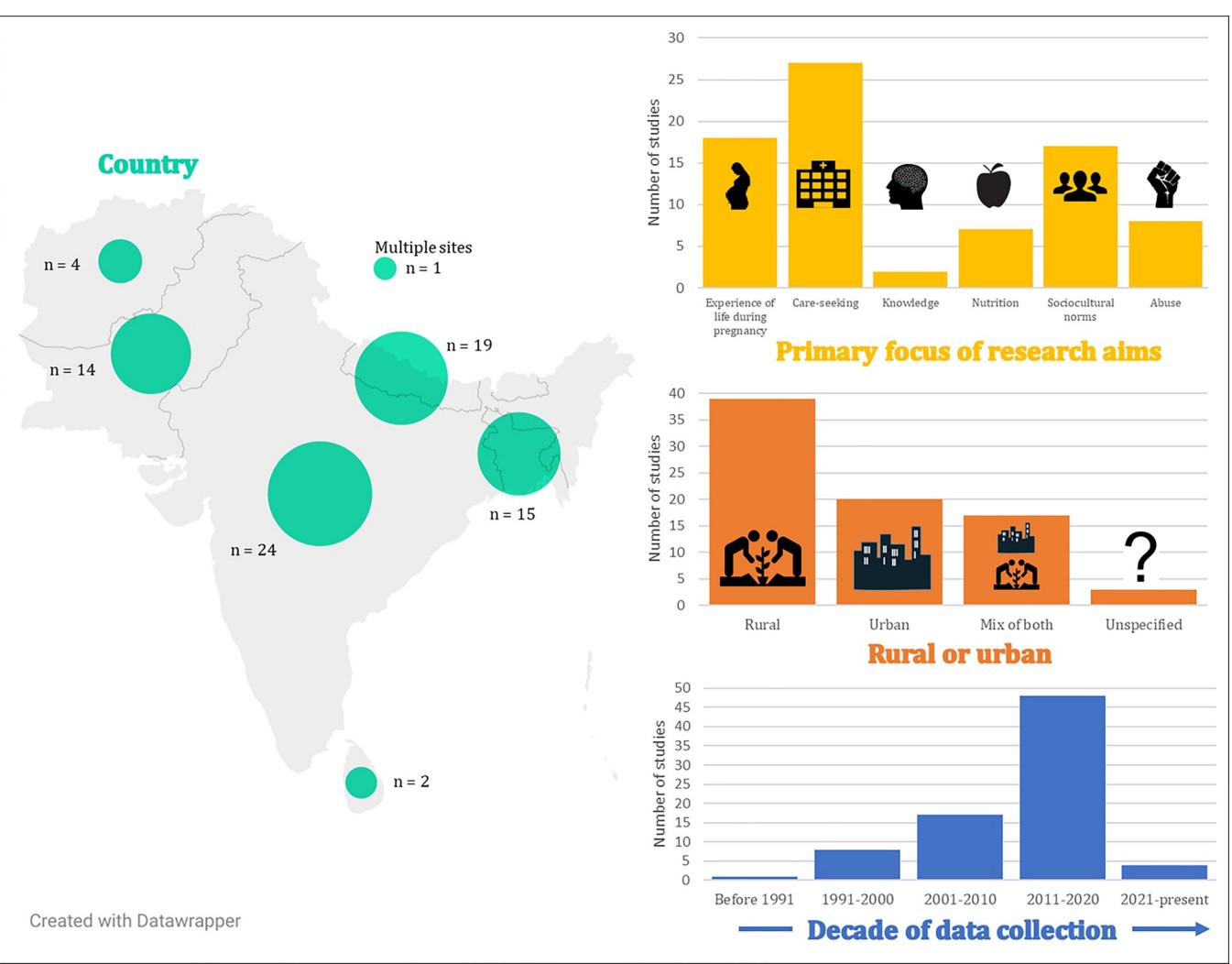

**Fig 3. Summary of study characteristics of the included studies, including country, focus of research aims, location, and decade of data collection.** The map was produced using datawrapper.de and the icons included in the graphs obtained from openclipart.org/. The authors hold the full copyright for these images.

girls to make decisions during pregnancy. Young pregnant women and girls felt unable to make decisions in several domains during pregnancy, including what they ate, what work they undertook, who they interacted with, when to seek care, and which care they received [45, 46, 49, 56, 73, 75, 81, 83, 96]. Instead, decisions were made collectively by the household, with their mother-in-law generally having the final say in reproductive matters [44, 47, 49, 56, 57, 73, 85, 107]. Most commonly, their husbands were not very involved in decision-making relating to reproductive health due to norms preventing their involvement [44, 47, 51, 52, 61, 63, 75, 115], as explained by this author of a study in rural Pakistan:

> "*In joint families. . . a man is considered besharam (shameless) if he exhibits an 'excessive' interest in his pregnant wife*" Author, 2001 [75]

Women of all ages had their decision-making ability restricted by household hierarchies; however, restrictions were stricter for younger women and girls and those who were married

**Table 2. Summary of high relevance studies.**

| Author | Title | Country (Region) | Participants | Methods | Outcome/area of focus |
|---|---|---|---|---|---|
| Rajbanshi et al. (2021) | Perceptions of good-quality antenatal care and birthing services among postpartum women in Nepal | Nepal (Morang) | 14 adolescents with a high-risk pregnancy who did not comply with hospital referrals: pregnant or within 42 days of birth | IDI *Thematic analysis, within behaviour change theory* | Reasons for non-adherence to hospital referral |
| Maharjan et al. (2019) | Factors influencing the use of reproductive health care services among married adolescent girls in Dang District, Nepal: A qualitative study | Nepal (Dang) | IDI: 14 Adolescent girls unknown # FGD (6-8/group) 10 KII | IDI, KII, FGD *Systematic text condensation, no theoretical framework presented* | Knowledge of health care-seeking practices among married adolescent girls |
| Paudel et al. (2018) | A qualitative study about the gendered experiences of motherhood and perinatal mortality in mountain villages of Nepal: Implications for improving perinatal survival | Nepal (Mugu) | 63 IDI: 42 women and families, 10 nurses/auxiliary nurses, 2 FCHV, 2 support staff, 1 auxiliary health worker, 2 traditional healers, 4 stakeholders (2 journalists and 2 activists) | IDI *Thematic analysis, sociocultural framework* | Experiences and beliefs of perinatal death |
| Shahabuddin et al. (2019) | Maternal health care-seeking behaviour of married adolescent girls: A prospective qualitative study in Banke District, Nepal | Nepal (Banke) | 52 IDI: 22 married pregnant adolescents, 10 married non-pregnant adolescents, 7 FCHV, 1 Government health officer, 1 health post worker, 1 medical doctor at hospital | IDI and FGD *Socioecological framework guided analysis* | Adolescent use of maternal health services |
| Cameron (1998) | On the edge of the auspicious: The practice and meaning of gender and caste in rural Nepal's low-caste households and society | Nepal (Bajhang) | Unknown # life histories: Women of different castes, including low-caste specialists | IDI for life histories, ethnographic observations, and engagement *Narrative analysis supplemented with cultural texts, theoretical framework centring caste and gender* | How gender and caste intersect in social experiences |
| Pike et al. (2021) | Family influences on health and nutrition practices of pregnant adolescents in Bangladesh | Bangladesh (urban slum in Dhaka and rural Rangpur) | 192 IDI: 96 pregnant adolescents or with <1 y/o, 64 family members, 32 healthcare providers | IDI *Identifying themes and gaps within socioecological framework* | Experiences of pregnant adolescents (ANC, nutrition, role of family) |
| Samandari et al. (2020) | Understanding individual, family and community perspectives on delaying early birth among adolescent girls: Findings from a formative evaluation in rural Bangladesh | Bangladesh (Kurigram Sadar and Rajarhat, Kurigram) | 20 unmarried girls <20yr, 21 newly married girls <20yr (within 1yr), 14 husbands, 47 influential adults, 15 community leaders, 10 community health providers | IDI *Thematic analysis, relating to theory of change* | Social norms and barriers/facilitators to delaying first birth |
| Shahabuddin et al. (2017) | Exploring maternal health care-seeking behaviour of married adolescent girls in Bangladesh: A social-ecological approach | Bangladesh (Rangpur) | IDI: 25 married pregnant adolescents, 10 married non-pregnant adolescents 3 FGD: 1 with 6 CHW, 1 with 7 community members, 1 with 6 MIL 4KII: 1 government officer, 2 NGO officials, 1 doctor | IDI, FGD and KII *Socioecological framework guided analysis* | Married adolescents' knowledge, perception, and use of reproductive and maternal health services |
| Rashid (2011) | Human rights and reproductive health: Political realities and pragmatic choices for married adolescent women living in urban slums, Bangladesh | Bangladesh (slum in Dhaka) | >60 IDI: 50 married adolescent girls, 12 husbands, additional family members 8 case studies: married adolescent girls | Ethnographic interviews, observations and interactions *Phenomenological approach* | How slum life impacts women's reproductive behaviour |
| Rashid (2006) | Emerging Changes in Reproductive Behaviour among Married Adolescent Girls in an Urban Slum in Dhaka, Bangladesh | Bangladesh (slum of Dhaka) | >60 IDI: 50 married adolescent girls, 12 husbands, additional family members 8 case studies: married adolescent girls 19 IDI: 6 slum leaders, 8 NGO workers, 1 clinic paramedic, 1 religious leader, 3 health workers | Ethnographic interviews, observations and interactions *Phenomenological approach* | Life histories and reproductive health histories |

*(Continued)*

**Table 2.** (Continued)

| Author | Title | Country (Region) | Participants | Methods | Outcome/area of focus |
|---|---|---|---|---|---|
| Schuler et al. (2006) | The timing of marriage and childbearing among rural families in Bangladesh: Choosing between competing risks | Bangladesh (villages in Rangpur and Magura) | 85 IDI: 20 men, 65 girls/women 24 FGD (unknown #/group): 4 groups | IDI and FGD *Grounded theory* | Norms, practices and decision-making related to marriage and pregnancy timing |
| Kamran et al. (2019) | Situation Analysis of Reproductive Health of Adolescents and Youth in Pakistan | Pakistan (Peri-urban areas in Islamabad, Lahore, Karachi, Peshawar, and Quetta) | > 250 adolescent boys and girls: 40 IDI, 24 FGD (~9/group) total unknown # | FGD and IDI *Grounded theory* | Cultural norms around reproductive health of adolescents |
| Hamid et al. (2009) | Who am I? Where am I? Experiences of married young women in a slum in Islamabad, Pakistan | Pakistan (urban slums, Islamabad) | 10 IDI: married adolescent women (conducted over multiple visits) | Unstructured interviews, married life calendar Narrative structuring, *Bronfenbrenner's ecological framework* | Knowledge of married life and reproduction at time of marriage |
| Perera et al. (2018) | 'When helpers hurt': Women's and midwives' stories of obstetric violence in state health institutions, Colombo district, Sri Lanka | Sri Lanka (Colombo) | 5 FGD: 28 PHM 6 FGD: 38 pregnant women (6-7/group) | FGD *Data organised within intersectionality framework* | Intersection of age, social class, and cultural background with obstetric violence |

Abbreviations: ANC; antenatal care, CHW; community health worker(s), FCHV; female community health volunteer(s), FGD; focus group discussion(s), IDI; in-depth interview(s), KII; key-informant interview(s), MIL; mother(s)-in-law, NGO; non-governmental organisation; PHM; public health midwife(s), y/o; year old, yr; year(s), unknown #; unknown number.

early. Families considered younger girls less mature and those married early less knowledge-able [46, 49, 51, 52, 59, 77, 112, 113], justifying their increased control over decision-making, as expressed by this 26-year-old aunt:

"*You are of a young age, whereas we are elder and we know more*" Urban Bangladesh, 2019 [49]

*Impact on care-seeking.* In most regions, young pregnant women and girls were often excluded from decision-making about care-seeking during pregnancy as this was their

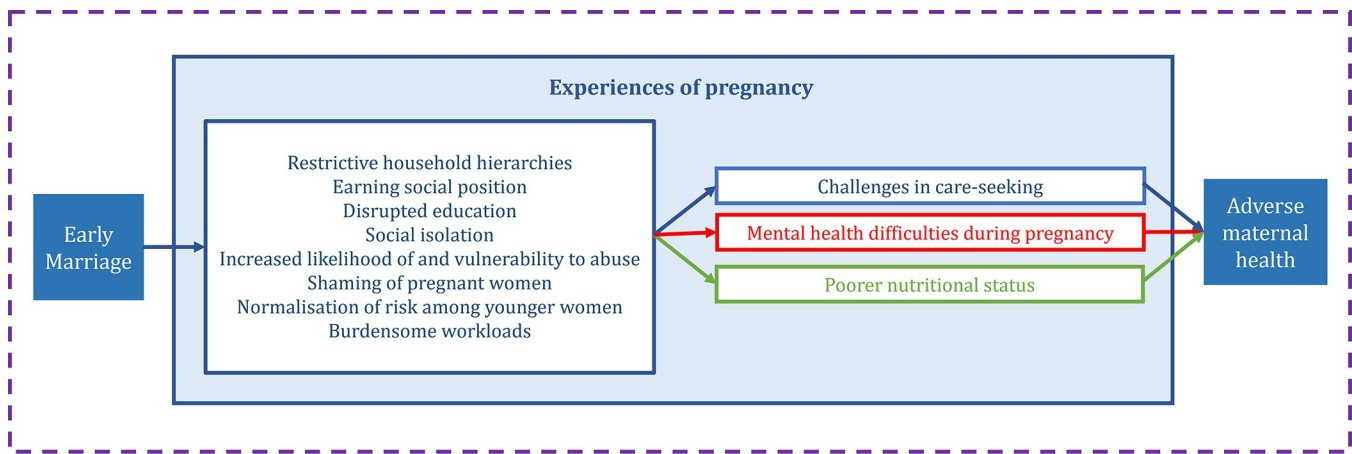

**Fig 4. Representation of the themes through which early marriage and early pregnancy influences the care-seeking, nutrition, and mental health of women and girls during pregnancy.**

mother-in-law's domain [46, 47, 56, 57, 104, 113], as expressed by this adolescent mother discussing their decision to attend ANC services during pregnancy:

"*I have to stay quiet. . . I can't say anything, they are senior to me*" Nepal, 2006 [96]

Care-seeking preferences of household-heads were often prioritised, even when these conflicted with healthcare provider advice [44, 49, 56, 85]. In extreme cases, women were not listened to in critical situations, resulting in complications which may have been preventable [44, 47, 80, 84], as was the case with this 17-year-old mother:

"*I knew that my condition was very serious, and everyone kept on telling me to try having the baby at home. I was trying, and I knew I couldn't try anymore, but the others didn't understand how serious it was.*" Rural Bangladesh, 2009 [84]

We identified two ways in which household hierarchies affected care-seeking among younger women and girls and those married early: i) younger women and girls felt less able to share their concerns and participate in care-seeking decision-making due to their lack of agency or shyness [52, 59, 62, 75, 80, 85, 93, 96, 113], and ii) household members were less likely to listen to their care-seeking requests due to a disbelief or distrust in younger daughters-in-law [44, 47, 49, 54, 56, 57, 73, 84].

However, there were regional differences. For example, the mothers-in-law of pregnant women and girls in Central and Southern India and Sri Lanka appeared less likely to enforce their beliefs [104, 117], as expressed by this mother-in-law from a study undertaken in South India:

"*Now, I don't stop my kids from going to the hospital for delivery but I think it is unnecessary*" 2021 [117]

*Impact on mental health*. Having their decision-making restricted during pregnancy was a major source of stress for young pregnant women and girls and those married early, as any conflicts with household heads prevented them from acting in line with their interests [51, 61, 63, 70, 84, 91, 93, 113]. Women were frustrated at being perceived as uneducated by household members who excluded them from decision-making and by healthcare providers who treated them with disrespect [49, 51, 93].

Pregnant women and girls were fearful of experiencing complications during pregnancy and felt helpless due to their lack of control [60, 84, 85, 91]. This is highlighted by this young woman reflecting on her experience of delivery:

"*My husband and his mother did not agree with the recommendations of 'dai' to take me to the hospital. . . I was pretty afraid, but I had to stay quiet due to the pressure of my husband and his family*" Rural Pakistan, 2013 [85]

Women and girls reported that navigating hierarchies in their new household during adolescence and while pregnant was overwhelming [51, 58, 83], such as this girl who married at age 12:

"*I was married off at a very early age, brought into a new family, I had to adjust with a new family and with my husband. Everything was haphazard and in the middle of all those changes I became a mother too. It was all very overwhelming*" Rural Bangladesh, 2013–14 [58]

*Impact on nutrition.* Young pregnant women and girls often relied on other household members to satisfy their nutritional needs; household members advised them what to eat, bought them food or provided them with money to buy food, decided on the allocation of food between family members, and influenced when they were able to rest during pregnancy [49, 60, 64, 71, 91]. Food allocation was negatively affected by norms around women eating last and sacrificing their share for their family [49, 64, 91], as was the case with this woman with a young child:

"*Who will cook again*? *My stomach is full if my family's stomach is full. I don't get hungry*" Nepal Terai, 2019 [91]

**Girls being pressured into pregnancy to earn social position within their household.** Women and girls felt pressure to increase their social position through marriage and delivering a baby, preferably a son [45–47, 52–54, 63, 70, 90, 94, 108]. These norms drove early marriage and repeated pregnancies, as explained by this woman who lost her baby at age 15 when asked why she got pregnant repeatedly at such a young age:

"*I know it is not good for my health. But my parents-in-law said to me that I should produce at least one child. Otherwise, they would bring a co-wife to my husband.*" Rural Nepal, 1998 [90]

Women and girls with poor marital relationships, common among those married early, felt these pressures more strongly as having a child was often viewed as a way to strengthen relationships [47, 52, 53, 55, 90, 93, 119]. Young and newly-married daughters-in-law were generally less trusted by their marital family, intensifying these pressures [45, 46, 52, 55, 85, 93, 108], as explained by this case study of a woman married at age 14:

"*My mother-in-law says I am not innocent and my husband says that I have to learn to be obedient and submissive*" Urban Pakistan [45]

*Impact on care-seeking.* The needs of young women and girls who had not yet 'earned' their social position were often not taken seriously by household members, particularly during their first pregnancy [52, 54]. Family members often did not trust women and girls when they had health complaints [47, 53, 63, 83], as reported by this teacher:

"*When she tells about her illness, the mother-in-law often does not trust; mother-in-law thinks that she is telling a lie (Nauragareko)*" Rural Nepal, 2015 [47]

*Impact on mental health.* The pressures felt by young girls increased their anxiety [44, 45, 47, 70, 73, 74, 94, 108], as explained by this author reflecting on the mental health of early married women in Northern India:

'*Such experiences were also recounted by other women and that it had caused them sadness and anxiety due to the immense social and familial pressure on them to prove their fertility and produce children soon after marriage.*' 2015–7 [94]

Women and girls were commonly blamed if they could not conceive or had experienced miscarriages, and families often provided threats such as divorce or a co-wife [46, 47, 52, 53, 55, 90, 93, 107, 119]. Adjusting to and living with these pressures was particularly upsetting for

younger women and girls, who were grappling with establishing their identity during adolescence [45, 83].

*Impact on nutrition.* Some pregnant women and girls saw fasting as a way of pleasing their in-laws and improving their chances of having a son [108].

**Early marriage affecting knowledge of pregnancy needs.** Girls usually left school before or shortly after marriage, negatively affecting their knowledge of reproductive and maternal health which was often taught in school [45, 46, 48, 49, 51, 54–56, 58, 60, 69, 80, 100, 109, 113, 115]. This 17-year-old mother expressed how her lack of education negatively affected pregnancy preparedness:

"*Who would do it if one knew everything? If I had known everything, I would not be pregnant*" Rural Nepal, 2015 [46]

However, even when women and girls were educated and understood their needs during pregnancy, they were often not able to raise their needs to household decision-makers, due to shyness, lack of agency, feeling overwhelmed, and/or thinking that they would not be listened to [46, 57, 70, 83, 93, 120]. Often, even girls who spoke out were not listened to [70, 73, 97, 108], such as this girl from a study in rural Uttar Pradesh who had married at age 13:

"*I understand the importance of doctors. I wanted to deliver my first child in the hospital, but my mother-in-law did not allow me to go*" Rural India [73]

*Impact on care-seeking.* Younger and less educated women were often less aware of warning-signs during pregnancy [17, 46, 49, 50, 55, 58, 80, 92, 106, 109]. Some women did not realise they were in labour as they did not recognise their labour pains and relied too heavily on their delivery date (which are known to be inaccurate due to recall error) [51, 58, 62, 69, 80, 115], such as this woman who experienced a neonatal death during adolescence:

"*At home, I was in labour pain for three days. It contracts and leaves. I thought the date had not yet approached. That's why I didn't even tell my husband.*" Nepal, 2019–2020 [51]

When accessing ANC services, uneducated women were often scolded by providers for their lack of knowledge, negatively affecting experiences and future care-seeking as a result [46, 49, 56, 80].

Households often prioritised the opinions of older female relatives over the perspectives of pregnant women, even when pregnant women were educated or their relatives' opinions conflicted with healthcare provider advice [49–51, 56, 69, 76, 85, 93, 97]. Healthcare providers considered this a barrier to the care they provide [49, 56, 113], as expressed by this nurse:

"*Family members are superstitious and tell, 'this is nonsense what the doctor says. Follow what we say.'*" Urban Bangladesh, 2014–15 [56]

*Impact on mental health.* Young women and girls and those married early acknowledged their lack of knowledge and expressed a sense of helplessness [49, 58, 60, 65, 88, 93, 107], such as this 15-year-old pregnant girl:

"T*hose who are underage. . . we understand less, or, we do not have the ability to understand*" Rural Bangladesh, 2019 [49]

Furthermore, large education gaps were common among couples who married early, which negatively impacted spousal communication and therefore their mental health during pregnancy [46, 58, 60, 108, 112].

*Impact on nutrition.* Less educated girls were less informed of their increased nutritional requirements during pregnancy [49, 50, 64, 108], allowing for misinformation to be perpetuated regarding food to be avoided and the benefits of physical labour during pregnancy [49, 50, 68, 90].

**Girls feeling isolated during pregnancy.**   Married women and girls commonly faced restrictions to their social interactions and independence. Studies from Northern India, Nepal Terai, Pakistan, and Bangladesh reported that girls were often prohibited from leaving the house unaccompanied [54, 74, 76, 81, 85, 91, 96]. This was not reported by girls in Southern India, although they still faced restrictions on their social interactions and behaviours [82, 93]. These restrictions increased during pregnancy due to the associated shame and stigma [49, 54, 59, 81, 82, 91, 104].

While most women experienced some level of restriction on their social behaviour, these were often tighter for younger women who were considered more vulnerable to influence [45, 51, 54, 70, 76, 82, 88, 91, 112], as expressed by this married adolescent reflecting on her path to early motherhood:

"*I do not go anywhere, even places near my house, because my mother-in-law suspects that if I talk to anybody, I might be badly influenced by them.*" Rural Bangladesh, 2017 [54]

Younger women found the navigation of relationships within their marital household difficult due to their shyness [45, 51, 76, 80, 87, 112, 120]. Furthermore, women who married early were often much younger and less educated than their husbands, negatively affecting how accepted they felt in their marital household [46, 58, 60, 108, 112].

Women and girls who had a close relationship with their natal family coped better with tensions within their marital household, partly because natal family relationships affected spouse choice but also because their natal household provided a place to visit during times of stress or in cases of abuse [45, 54, 63, 65, 104], as for this 19-year-old pregnant girl in an abusive relationship:

"*So, whenever we have a big argument I visit my maternal home*" Urban Nepal, 2013 [63]

*Impact on care-seeking.* In studies from Northern India, Nepal Terai, Pakistan, and Bangladesh, women were often not permitted to leave home unaccompanied, and therefore relied on their relatives to attend healthcare appointments with them [54, 74, 76, 81, 85, 91, 96]. If their relatives were unsupportive of care-seeking, this negatively impacted their ability to seek care [61, 93, 96, 120], as with this pregnant adolescent:

"*ANC check-up is not easy for me. I have problems at home. [My in-laws] won't allow me to go outside much. I cannot go for ANC check-up on my wishes. They said that I do not need to go again.*" Nepal, 2006 [96]

However, some participants were allowed to visit facilities with neighbours or relatives of a similar age, overcoming some of the restrictions to unaccompanied mobility [49, 56, 57, 87, 97], as this teenage mother who lived far from the ANC centre reflected on:

"*I had a pregnant neighbour with close delivery dates. We would go together*" Rural Nepal, 2014–15 [57]

*Impact on mental health*. Transitioning directly from relative freedom during childhood to restrictions in marriage was difficult [49, 54, 70, 81, 87, 88, 112], as explained by this pregnant adolescent when explaining how pregnancy has impacted her daily life:

"*I was able to live independently. Now I walk less, hang around less; I have to sit at home, can't go anywhere outside. Can't live willingly.*" Urban Bangladesh, 2019 [49]

Young women and girls commonly reported feeling lonely and helpless during pregnancy due to a lack of social support [45, 47, 49, 51, 61, 70, 81, 88, 112], such as this 17-year-old woman who was continuing an unwanted pregnancy against her wishes:

"*I am feeling lonely and helpless. . . I am so anxious about it. I am feeling sick. I have fear. I am worried.*" Rural Nepal, 2010 [70]

*Impact on nutrition*. Young women and girls were generally unable to travel to local markets to purchase nutritious food, needing to rely on other household members to do so [49, 91].

**Increased threat of abuse for adolescents.**   Emotional, sexual, and physical abuse was normalised among married women and girls across South Asia [45–47, 52, 53, 63, 65, 67, 73, 78, 79, 81, 89, 94, 110], to the extent that this early-married woman living in an urban slum in Pakistan viewed sexual abuse as a symbol of love:

"*Unwanted sex is also a symbol of love. It is a way to resolve the dispute between husband and wife.*" 2013 [78]

Girls were commonly subjected to abuse from multiple household members, increasing their vulnerability [45–47, 65, 89], as in this case study of an early-married woman:

"*He complained to his mother who told him to tie me up and have sexual activity with me anyway.*" Urban Pakistan [45]

For some, abuse intensified during pregnancy [52, 63, 65, 78, 79, 107] and injuries sustained from violent physical and sexual acts were suspected to have caused women and girls to miscarry [67, 78, 79, 81, 89, 110], as was the case for this early-married pregnant woman:

"*Even last night they beat me, and I am bleeding. I am four months pregnant and I am bleeding. . . he uses his hands and he throws me here and there against the walls*" Nepal [67]

Younger and early married women were more likely to be subjected to abuse [48, 65, 67, 73, 79, 81, 83, 89, 94, 110]. Having a much older or more educated husband, common among early married women, intensified abuse due to poor spousal communication [65, 73]. Girls in love marriages were also more likely to be subjected to abuse, as their in-laws were less likely to be accepting of their marriage [52, 53, 63]. Younger girls were also considered less physically and emotionally strong, and therefore less able to endure abuse [55, 73, 89].

*Impact on care-seeking*. Women and girls who were subjected to abuse may have been less likely to seek care; some delayed telling their household members about their pregnancy due

to fear of retaliatory abuse [73], while the families of others prevented care-seeking out of fear that they would speak out [89], as reported by the author of this study in Karachi, Pakistan:

> "*Her husband agreed to hospitalization, but he threatened her with divorce if she divulged any information about his violent behaviour to the hospital staff*" [89]

Girls also frequently experienced verbal abuse and neglect from healthcare providers, affecting the quality of care provided and future care-seeking [45, 46, 48, 49, 53, 57, 69, 71, 89, 95, 101, 113]. They were scolded by providers for marrying early, becoming pregnant early, having repeated pregnancies, and being less likely to follow their advice [46, 48, 49, 52, 53, 57, 62, 69, 95, 113]. Women and girls felt helpless in these situations, as expressed by this adolescent mother:

> "*I was left alone in the stirrups legs up. I was screaming and nobody came.*" Urban Afghanistan, 1996 [69]

*Impact on mental health*. Being abused made women and girls feel scared and unsafe, negatively impacting their mental health [45, 52, 53, 63, 65, 79, 81, 89, 94], as expressed by this 17-year-old pregnant girl:

> "*He beats me till he cools down and blames my maternal home for giving birth to me. I feel so sad and I regret marrying him (tears in her eyes)*" Urban Nepal, 2013 [63]

Younger women were more emotionally vulnerable to these effects, as reported by this author reflecting on healthcare-provider violence:

> "*Young women having their first babies were particularly distressed but their cries for help, for their mothers or for God were largely ignored.*" Urban Afghanistan, 2010–12 [101]

Women and girls who were not able to return to their natal household suffered in particular, as this was a common fall-back option [46, 47, 63, 65, 89, 94, 110].

*Impact on nutrition*. Women and girls often felt sick and did not feel like eating due to stress caused by the abuse [63, 70]. Some women had their food withheld by abusers, severely impacting their nutritional status [52, 73, 89], such as this woman who married at age 13 years:

> "*I was young and weak, I opposed him, he used to beat me, stopped giving me food, even tortured me during my periods. . . I wish I was never born*" Rural India [73]

Furthermore, women and girls were commonly threatened with abuse if they did not work hard enough, causing them to become overworked and exhausted during pregnancy [44–46, 59, 94].

**Pregnancy being a time of shame.** Across South Asia, women and girls felt conflicted as they transitioned from their unmarried state, when their bodies represented modesty and chastity, to pregnancy, when their bodies represented sexual activity [44, 56, 57, 72–74, 76, 83, 89, 90, 99, 104]. Studies from Pakistan, Bangladesh, Northern India and Nepal reported that pregnant women and girls often had their movements restricted by their families as a result [56, 57, 73, 76, 85]. This author explained how in Pakistan, a pregnant belly represents sexual activity:

"*Pregnancy, an obvious manifestation of sexual activity, is associated with 'sharam' [shame]. Pregnant women should avoid public space*" Rural Pakistan, 2001 [76]

Related to this, pregnant women and girls generally felt embarrassed and ashamed when discussing their pregnancy [17, 44, 72, 83, 89, 91, 99, 104, 115]. Women who were younger, less educated, and had a faster transition from childhood to pregnancy tended to be more shy and embarrassed when discussing their pregnancy [44, 46, 62, 72, 83, 91, 99, 120].

*Impact on care-seeking.* Women and girls frequently felt uncomfortable sharing pregnancy-related health concerns with their in-laws, meaning they delayed care-seeking [57, 89, 91, 104, 120], as with this young daughter-in-law living in rural Maharashtra:

"*I had red discharge and pain in my abdomen. I was ashamed of telling (my mother-in-law).*" 1996 [103]

Pregnant women and girls often felt ashamed to seek care [46, 56, 57, 62, 73, 90, 96], particularly from male doctors, as reported by this pregnant girl aged 15:

"*I will be ashamed. The doctors see your body. A lot of people see your body. That's why I didn't feel like going to the medical.*" Rural Bangladesh, 2014 [56]

*Impact on mental health.* Women and girls found it difficult to manage their conflicting and shameful feelings during pregnancy, particularly as they felt unable to share their concerns with others [45, 58, 94, 104, 108, 113], as represented by this case study of an early married woman in Pakistan:

"*I am not innocent and my husband says that I have to learn to be obedient and submissive. I am not allowed to leave home unaccompanied. I worry for myself. Where is my home and who am I?*" [45]

**Risk being normalised among younger pregnant women.** Across South Asia, younger girls were thought to have an increased risk of experiencing complications during pregnancy and delivery due to their physical immaturity [21, 48–50, 54, 59, 60, 65, 68, 71, 74, 85, 88, 95, 98, 100, 102, 105, 109, 116, 119, 121]. This was attributed to early marriage [47, 51, 55, 66, 77, 88, 110].

Therefore, younger women experienced pregnancy with an expectation and normalisation of risk based on the experiences of friends and relatives, while feeling powerless to act to improve their outcomes [44, 47, 51, 60, 73]. This expectation is expressed by this young pregnant woman in rural Nepal:

"*Well, I think I am going to die anyway, so what does it matter if I have a son? All I wish is that it lives. Will it live or die? Nobody knows.*" 1988/89 [44]

*Impact on care-seeking.* Women and girls often delayed care seeking as a result of this normalisation of risk, as they considered pregnancy to be a condition needing care only in complicated cases [46, 47, 49, 56, 58, 62, 71, 74, 77, 80, 85], as explained by this young married woman:

"*As I had a normal delivery, I never visited the doctor because there were no complications.*" Urban India, 2020–21 [71]

Many considered younger women to have an increased risk of needing a caesarean-section, particularly in Bangladesh [49, 55, 56, 60]. This pregnant adolescent explained that she planned to deliver at home for this reason:

*"There are a lot of problems if you go to the hospital, if you do a C-section there is a problem in moving. There is a problem in eating, for three to four months you can't do any heavy work"* Rural Bangladesh, 2014–15 [56]

*Impact on mental health*. Stories of adverse outcomes among friends and relatives were upsetting for young girls [44, 60, 70, 71, 73, 81, 88]. Coupled with the fact that they were generally unable to make decisions to improve outcomes, women and girls often felt anxious about this normalisation of risk [50, 60, 70, 71], as with this young mother whose friend had died during delivery:

*"I was very afraid during delivery because I thought I would die. . . One of my friend expired (died) during delivery at the age of 16. . ."* Urban Nepal, 2015 [60]

**Conflicting reproductive, domestic, and economic roles.** Many women and girls faced demanding domestic workloads within their marital households, often with additional responsibilities from agricultural and income generation work [44–47, 49, 52, 55, 59, 64, 68, 80, 83, 90, 94, 108, 110, 112, 120]. The extent to which their health and wellbeing during pregnancy was affected depended on the type of work and the extent to which their workload expectations aligned with their household's expectations [44, 46, 47, 49, 52, 53, 55, 58, 110].

In rural environments, women and girls were often engaged in agricultural work, which families generally expected to continue throughout pregnancy [44, 46, 55, 59, 68, 83, 90]. This work was often dangerous, and accidents could cause miscarriages or stillbirths [44, 46, 47], as this lady health worker described:

*"She had gone to bring fodder from the* forest. *She slipped on the road, hurt her abdomen. She had bleeding and fainted. . . Fortunately, the mother survived; however, the baby died"* Rural Nepal, 2015 [47]

In urban environments, women were often involved in income generating work [52, 53, 55, 58]. Pregnancy was often delayed to prioritise income generation, or women were expected to give up work to start a family [52, 53, 55, 71, 110]. The expectation depended on the preferences of their in-laws, as expressed by this girl who married at age 14:

*"(My husband) really wanted a baby. I was working in the garments. I wanted to work for longer, save up some money and then have a child. . . My mother convinced me to have a baby to make him happy."* Urban Bangladesh, 2001–03 [52]

Younger women and girls and those married early felt less able to negotiate with their in-laws regarding their workload [45–47, 49, 52, 53, 58, 64, 80, 90, 94, 112]. Younger mothers were also considered more vulnerable to exhaustion due to their physical immaturity [47, 50, 59, 71, 94, 108]. The families of women and girls who married early were also less likely to have invested in their education, limiting their employment prospects [45, 55, 58, 83].

*Impact on care-seeking*. Heavy workloads prevented women from seeking care [46, 47, 56], as was the case with this early married woman who delivered at home due to workload demands:

"*I also delivered the baby at home. I was working all morning and in the evening. . . Nobody was aware at home that I was having the labour pain.*" Rural Nepal, 2015 [46]

*Impact on mental health*. Women and girls were anxious due to their demanding work-loads, and for the potential that pregnancy complications may prevent them from working after childbirth [47, 52, 58, 64, 94, 112]. Heavy workloads also prevented them from visiting their natal family [80, 110].

However, work outside the home also provided autonomy, freedom of movement, a social network outside the household, and some economic independence [52, 53, 55, 58, 75].

*Impact on nutrition*. Pregnant women and their families acknowledged the risks associated with demanding physical labour during pregnancy, such as exhaustion and anaemia [46, 47, 71, 90]. However, some believed that heavy physical labour was beneficial during pregnancy [68, 90], as expressed by this husband whose wife had recently delivered:

"Pregnant women do every kind of work. The harder your work during pregnancy, the eas-ier it will be for delivery" Husband, rural Nepal [90]

## Discussion

This review found that several interconnected factors shape experiences of pregnancy follow-ing early marriage in South Asia. The two most important factors we identified were the social position of women and girls within their household, and the quality of their relationships within and outside of their home. By presenting how these interpersonal factors connect with other factors, such as education, experiences of abuse, and burdensome workloads, we provide a new perspective which highlights the importance of households and communities listening to and trusting women and girls.

Additional themes we found in this review highlight how, due to their young age and lack of education, respectively, younger and early married women felt more shy and less empow-ered and were considered less knowledgeable. These were used as justifications for households maintaining control over decision-making. Early married women struggled with the restric-tions placed on their social interactions, as this prevented them from forming and maintaining relationships and exacerbated the transition from childhood to married life. Younger women were considered to have an increased risk of pregnancy complications due to their physical immaturity, leading to an expectation of negative outcomes which increased anxiety and nega-tively affected care-seeking behaviours.

Our findings should be considered within the limitations of this review. While we searched for studies from across South Asia, most studies were from India, Nepal, Bangladesh, and Paki-stan, with few from Sri Lanka or Afghanistan, and none from Bhutan or the Maldives. The lack of perspectives from these regions limited our ability to draw comparisons on experiences of pregnancy between regions. Furthermore, findings may not apply to all regions within South Asia due to sociocultural and demographic differences between countries and sub-national regions. Furthermore, context-specific evidence may have been decontextualised while synthesising findings from studies across South Asia. As we aimed to identify potential pathways through which early marriage impacts maternal health, we may have had a negative bias when synthesising findings. However, we sought out contradictory evidence when devel-oping themes to challenge this bias. Furthermore, by excluding studies exploring the experi-ences of married women and girls outside of pregnancy, we exclude the potential to explore relevant experiences before pregnancy. However, inclusion of these studies would have made the scope of this review too broad. We were limited in the extent to which we could disentangle experiences of early marriage and early pregnancy, as early marriage was considered a pre-

requisite to early pregnancy and therefore, they were often considered together. Despite these limitations, this review provides new perspectives on the experiences of pregnancy following early marriage in South Asia. Of the three pathways we identified, the evidence base exploring care-seeking experiences was strongest, whereas qualitative evidence exploring experiences of mental health and nutrition was lacking.

Quantitative evidence from South Asia supports our finding that younger women are less involved than older women in decision-making during pregnancy [98, 124–127]. In this review, some girls reported actively diminishing their agency during household decision-making, suggesting they acknowledge the benefits of acting in line with rather than challenging hierarchies in decision-making within households [49, 85, 90]. This supports other conceptualisations of household bargaining in South Asia which highlight that an absence of protest does not reflect an absence of questioning [13, 128]. As household hierarchies frequently delayed or prevented care-seeking, we join others who call for an update to existing models which seek to understand care-seeking to incorporate the nuance of women's decision-making capabilities [129, 130]. We found that newly married women and girls in particular had their decision-making restricted, as they have not yet gained status through their reproductive role. However, quantitative evidence from Nepal and Bangladesh suggests that the household status of women and girls does not increase with time or with the delivery of a child [131–134]. These nuances are context-specific and difficult to measure, requiring research at a local level, which we were unable to undertake in this regional review [135]. Programs to improve experiences during pregnancy in South Asia must address household dynamics, emphasising the importance of families listening to and trusting women and girls and prioritising their needs.

Current conceptualisations of experiences of pregnancy following early marriage focus on the importance of education [136–140]. This is supported by quantitative evidence from South Asia which has found associations between education, knowledge of reproductive health [19, 109–111], and decision-making [127, 131, 135, 141]. However, this review found that even educated girls were generally unable to speak up and be listened to within their households due to restrictive household hierarchies, meaning that education does not translate into improved experiences. This inconsistency may reflect the opposing mechanisms of caste and education; while women from disadvantaged caste groups are less educated, their households are described as more egalitarian meaning they may be more involved in decision-making [142, 143]. Efforts to improve the educational attainment of girls must be accompanied with efforts to shift household hierarchies and social norms for them to be able to enact learned knowledge.

Socioeconomic vulnerabilities are another commonly cited driver of early marriage and mediator of experience. Indeed, we found evidence to suggest that socioeconomic deprivation and unemployment limited the opportunities for women and girls outside of their marriage, making them less able to negotiate their needs within their household [52, 53, 71, 87]. Agarwal refers to this as a woman's 'fall-back position', associating lower fall-back positions (fewer options if the marriage fails) with lower bargaining power within a household [128]. However, we also found that socioeconomic reasons were often given for not seeking care or following recommended nutritional practices, despite services and supplements being offered for free [44, 49, 56, 85, 90]. Inconsistencies within the quantitative literature on associations between socioeconomic status and pregnancy decision-making [131, 144, 145] may be explained by household dynamics, as sociodemographic factors are understood to be less important in contexts where women's agency is more restricted [146]. These conflicting findings highlight the importance of understanding the local contexts in which women and girls live when undertaking research and designing programs.

Another set of interwoven factors relates to the societal shaming of sexual activity and the restrictions placed on the social interactions of women and girls. Our findings are consistent with the quantitative literature which supports an association between early marriage, social restrictions, and reduced care-seeking [146–148]. However, the extent to which these factors influenced pregnancy experiences differed between regions. Further efforts to harness the potential of social groups to deconstruct the shameful connotations associated with pregnancy should be explored to improve care-seeking and socioemotional wellbeing.

While we identified few qualitative studies focusing primarily on mental health during pregnancy, each theme highlighted mechanisms through which early marriage affects mental health due to a lack of autonomy and control over their lives. This is supported by longitudinal evidence from Northern India, which found a reciprocal relationship between early marriage and mental health; girls who marry early reported worse mental health before marriage, which proceeded to deteriorate further after marriage [149]. However, quantitative evidence from Pakistan questions the assumed association between autonomy and mental health in South Asia, highlighting that in this context the promotion of self-denial and self-sacrifice changes people's conceptualisation of mental health relative to other settings, calling for context-specific indicators [150]. Further research considering how context- and pregnancy-specific factors intersect to influence the mental health of young and early married women is needed.

## Conclusion

We highlight the extent to which relationships within households and communities affect the health and wellbeing of early married women in South Asia. Individual-level factors, such as education and empowerment, can improve the knowledge of women and girls. However, translating this into improved care-seeking, mental health, and nutrition requires engagement at the household and community levels to ensure women and girls are being trusted and listened to.

## Supporting information

**S1 Appendix. Complete search strategies for each database: Ovid (MEDLINE, EMBASE, PsychINFO, and Global Health), Scopus, Global Index Medicus (WHO), CINAHL (EBSCO), Web of Science, and PROQUEST.**
(DOCX)

**S2 Appendix. Quality appraisal for high, medium, and low relevance studies, as determined using the Critical Appraisal Skills Programme (CASP) quality appraisal tool and the relevance of the study to the review question, organised by country and year of publication.**
(DOCX)

**S3 Appendix. Data extraction tables for medium and low relevance studies, organised by country and year of publication.**
(DOCX)

**S4 Appendix. Heat map of themes and subthemes, in which a darker colour representing a higher referenced theme, with supporting evidence.**
(DOCX)

**S5 Appendix. Completed PRISMA 2020 checklist addressing the reporting of review components in the title, abstract, introduction, methods, results and discussion sections of a**

**systematic review report.**
(DOCX)

## Acknowledgments

We thank Heather Chesters, the deputy librarian at UCL Institute for Global Health, for their advice when conducting the searches for this review.

## Author Contributions

**Conceptualization:** Faith A. Miller.

**Data curation:** Faith A. Miller, Sophiya Dulal, Anjana Rai.

**Formal analysis:** Faith A. Miller, Sophiya Dulal, Anjana Rai, Helen Harris-Fry, Naomi M. Saville.

**Funding acquisition:** Faith A. Miller.

**Investigation:** Faith A. Miller, Sophiya Dulal, Anjana Rai, Lu Gram, Helen Harris-Fry, Naomi M. Saville.

**Methodology:** Faith A. Miller.

**Project administration:** Faith A. Miller.

**Resources:** Faith A. Miller.

**Software:** Faith A. Miller.

**Supervision:** Lu Gram, Helen Harris-Fry, Naomi M. Saville.

**Validation:** Faith A. Miller.

**Visualization:** Faith A. Miller.

**Writing – original draft:** Faith A. Miller.

**Writing – review & editing:** Sophiya Dulal, Anjana Rai, Lu Gram, Helen Harris-Fry, Naomi M. Saville.

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
