## [Decision Letter · Decision Letter 0]

21 Jun 2023

PGPH-D-23-00724

“Can’t live willingly”: A thematic synthesis of qualitative evidence exploring how early marriage and early pregnancy affect experiences of pregnancy in South Asia

Dear Dr. Miller,

Thank you for submitting your manuscript to PLOS Global Public Health. After careful consideration, we feel that it has merit but does not fully meet PLOS Global Public Health’s publication criteria as it currently stands. Therefore, we invite you to submit a revised version of the manuscript that addresses the points raised during the review process.

Editor comments: 

The reviewers and I found the manuscript to be well-written. It offers a helpful synthesis on the qualitative literature on early marriage and pregnancy in South Asia, and indicates important future directions for further qualitative work.In addition to the reviewers' suggestions below, I would encourage the authors to discuss the limitations of their approach a bit more in the Discussion. They point to some potential limitations in terms of conclusions that can be drawn from the analysis, but it would also be helpful to identify potential limitations in the methodological approach. For example, as noted by one reviewer, there are some very old papers included in this review, and it isn't clear what the date range was. In addition, a focus on peer-reviewed manuscripts will leave out a lot of ethnographic work presented in book chapters and books, which may offer more detail on both methods and results not possible in a shorter manuscript. There may be others, but these occurred to me as I read the manuscript. Minor comment: In the discussion, the authors refer to a study from Pakistan that discusses mental health, but state that this paper was not included in the review. It is a little unclear as to why this paper was not included in the review? 

We look forward to receiving your revised manuscript.

Kind regards,

Marie A. Brault, PhD

Academic Editor

Journal Requirements:

1. We have noticed that you have uploaded Supporting Information files, but you have not included a list of legends. Please add a full list of legends for your Supporting Information files after the references list. 

Additional Editor Comments (if provided):

Reviewers' comments:

Reviewer's Responses to Questions

**Comments to the Author**

1. Does this manuscript meet PLOS Global Public Health’s publication criteria? Is the manuscript technically sound, and do the data support the conclusions? The manuscript must describe methodologically and ethically rigorous research with conclusions that are appropriately drawn based on the data presented.

Reviewer #1: Yes

Reviewer #2: Partly

2. Has the statistical analysis been performed appropriately and rigorously?

Reviewer #1: Yes

Reviewer #2: N/A

3. Have the authors made all data underlying the findings in their manuscript fully available (please refer to the Data Availability Statement at the start of the manuscript PDF file)?

Reviewer #1: Yes

Reviewer #2: Yes

4. Is the manuscript presented in an intelligible fashion and written in standard English?

Reviewer #1: Yes

Reviewer #2: Yes

5. Review Comments to the Author

Reviewer #1: This paper attempts to understand ‘how early marriage and early pregnancy interact in shaping pregnancy experiences’ by an inductive approach, namely thematic synthesis of literature from South Asia. It fills the gap in our understanding of mechanisms or pathways through which early marriage and early pregnancy affect maternal health outcomes. Outcome includes experiences of pregnancy or childbirth, including but not limited to nutrition, psychosocial health, care-seeking, and family relationships. The use of PICOS framework to outline eligibility criteria is apt. PRISMA flow diagram for review of screening has also been included. Based on the thematic analysis, the authors conclude that intra-household relationships and relationships within communities need to be considered when designing strategies to improve outcomes from early marriage and early pregnancy.

The paper is generally well written and reflects the meticulous efforts of the authors. It was part of the first author’s PhD thesis. I have also reviewed the protocol registered on PROSPERO.

I believe the paper will benefit from minor revisions. My specific comments are as follows.

Specific Comments

1. In the abstract as well as the main text, the meaning of ‘household hierarchies’ as in ‘restrictive household hierarchies’ and ‘family hierarchy’ may be qualified to avoid ambiguities in interpretation. Are these hierarchies nested in social norms or they can independently influence the outcomes being studied? To what extent are the interpretations peculiar to specific household structures?

2. It is important to note that reviews of quantitative studies also acknowledge heterogeneity in the results of studies, as in the case of review of qualitative studies. However, epistemological contrasts may be indicated.

3. At p.4, the authors say “While South Asia is a culturally diverse region, represented by its linguistic, religious, ethnic, and geographical diversity, similar social structures, norms, and values exist which affect experiences of early marriage (20,22,23,26).” The authors may consider adding a line on considerable intra-regional heterogeneities at a sub-national level as well, which is alluded to later (p.13). This will strengthen their emphasis on ‘context-specific evidence’.

4. What are the range of dimensions for variation in qualitative synthesis? A succinct description of the same will help readers understand the pros and cons of alternative qualitative synthesis approaches.

Please provide citations for description of the search strategy. For example, why was forward and backward citation searching undertaken? Similarly, please provide some benefits of CASP tool for appraising study quality, and any caveats that one needs to be informed about.

5. At p.4, what was the motivation for: “Medical Subject Heading terms, and database-specific limiters, with no language or date restrictions.”

6. Why was an additional question on theoretical underpinnings included? (p.5)

7. Is the development of descriptive and analytical themes using coding related to reciprocal 'translation' as in meta-ethnographies (Noblit and Hare 1988)? Please briefly indicate.

8. What is the extent of dependency on the South-Asian co-authors for contextualization of findings? (p.6)

9. There’s a need for caution in generalization about intra-regional ‘North’ and ‘South’. Is the attribution on the basis of the individuals being a native of the geographical or cultural region or they happen to reside there? For example, what is the geographical reference for ‘mother-in-law from South India’? (p.13)

10. What could explain lack of high relevance studies undertaken in India, which has considerable quantitative evidence emerging on the topic? This is also surprising given the general increase in number of studies over time.

11. At p.11, what do the authors mean by ‘decisions’ in ‘household hierarchies limiting the ability of women and girls to make decisions’? There are several domains of decision making in a household context that the literature delves into, and it would be necessary to clarify the specific domain(s) of interest as decisions about care seeking during pregnancy follow in the subsequent paragraph.

12. What could explain the lack of emergence of positive narratives that have been associated with socio-economic transformation and improvement in health infrastructure as well as public RCH programs in general? This is critical from the perspective of the general improvement in educational attainment of girl child and women in South Asia over the years (briefly acknowledged at p.34). There are spatial and temporal exceptions including pre- and post-Taliban Afghanistan for instance, nonetheless. The authors acknowledge the negative bias (p.33) but to what extent is a consequence of the approach of synthesis can be commented upon.

13. The relevance of intersectionality presented in the paper (p.31) offers scope for critical interpretation of the findings presented prior to that section.

14. In the discussion section, it would be informative to clearly indicate what the authors consider as new perspectives on the experiences of early marriage in South Asia that the review offers.

15. In the concluding paragraph (p.35), it is not clear as to how the interventions at different levels (individual and community) connect with each other as was evident from the thematic analysis. Please clarify.

Reviewer #2: I wish to profusely thank you for the opportunity given me to review this manuscript titled ‘“Can’t live willingly” A thematic synthesis of qualitative evidence exploring how early marriage and early pregnancy affect experiences of pregnancy in South Asia” and in the end provide you with recommendations on the appropriateness or otherwise of the manuscript for publication. I diligently reviewed the manuscript, and I wish to commend the authors seriously for the efforts put in to develop this article and for their interest in publishing in the journal. The manuscript was well organized into the various sections of a scientific manuscript, and written in simple as well as clear text for easier communication of the readership community. Both full and short titles are appropriate and covers the scope of the study.

The manuscript is original article that is fairly well organized in a standard format for systematic reviews with sections on the abstract, introduction, methods, results, discussion, references, tables, figures and appendices. The research assessed the qualitative evidence on adverse experiences of pregnancy following early marriage or early pregnancy in South Asia, and revealed the mechanisms between early marriage and adverse pregnancy outcomes. However, the following corrections should be taken note of:

In line 21 remove during pregnancy because it opposes the previous words (younger maternal age), a woman first gets pregnant before becoming a mother.

Line 22, mention a few additional factors as the statement is incomplete.

Line 29, 79 begins the sentence and should be written in words.

Line 30, confirm that that the tool used is the Critical Appraisal Skills for Qualitative Study. If so, write out in full.

Line 32, experiences depicts both positive and negative. Hence use poor pregnancy experiences since your focus is on the negative aspect.

Line 40, Poor pregnancy experiences, not pregnancy experiences.

Line 48 and 49, include reference for South Asia early marriage statistics.

Line 67, add comma after over recent decades.

Line 76, show the direction of pregnancy experiences, you are interested in.

Line 77, show direction of maternal health outcomes.

Line 89, include in the S1 Appendix, the number of articles after each search.

Line 101, include reference of the exact Critical Appraisal Skills Program tools that was used.

Line 102, begin sentence in words, the figure can be written in bracket.

Line 125, write 24 in words because it begins the sentence and remove space in the word “were”.

To present quality evidence, I strongly suggest that the range of years should have been specified during the study selection to allow only papers relevant to the current times. For example, publications from year 2000 and till date. This can be seen in Line 242, the paper was published 30 years ago and is rather outdated compared to the recent publication in Line 200 which was published in year 2021. This also applies to Line 428 and Line 618, a 1996 and 1988 publication respectively.

Lastly, Line 714, this is more of a recommendation than a conclusion. Conclude on the study before suggesting recommendations.

6. PLOS authors have the option to publish the peer review history of their article (what does this mean?). If published, this will include your full peer review and any attached files.

**Do you want your identity to be public for this peer review?** For information about this choice, including consent withdrawal, please see our Privacy Policy.

Reviewer #1: **Yes: **Sarthak Gaurav

Reviewer #2: **Yes: **Andrew-Bassey, Uduak Ima

---

## [Decision Letter · Decision Letter 1]

15 Sep 2023

“Can’t live willingly”: A thematic synthesis of qualitative evidence exploring how early marriage and early pregnancy affect experiences of pregnancy in South Asia

PGPH-D-23-00724R1

Dear Miss Miller,

We are pleased to inform you that your manuscript '“Can’t live willingly”: A thematic synthesis of qualitative evidence exploring how early marriage and early pregnancy affect experiences of pregnancy in South Asia' has been provisionally accepted for publication in PLOS Global Public Health.

Best regards,

Marie A. Brault, PhD

Academic Editor

Reviewer Comments (if any, and for reference):

Reviewer's Responses to Questions

**Comments to the Author**

1. If the authors have adequately addressed your comments raised in a previous round of review and you feel that this manuscript is now acceptable for publication, you may indicate that here to bypass the “Comments to the Author” section, enter your conflict of interest statement in the “Confidential to Editor” section, and submit your "Accept" recommendation.

Reviewer #2: All comments have been addressed

2. Does this manuscript meet PLOS Global Public Health’s publication criteria? Is the manuscript technically sound, and do the data support the conclusions? The manuscript must describe methodologically and ethically rigorous research with conclusions that are appropriately drawn based on the data presented.

Reviewer #2: Yes

3. Has the statistical analysis been performed appropriately and rigorously?

Reviewer #2: N/A

4. Have the authors made all data underlying the findings in their manuscript fully available (please refer to the Data Availability Statement at the start of the manuscript PDF file)?

Reviewer #2: Yes

5. Is the manuscript presented in an intelligible fashion and written in standard English?

Reviewer #2: Yes

6. Review Comments to the Author

Reviewer #2: I commend the rigorous work the authors of this paper have done in keeping to detail. All comments that were initially suggested have been revised. Also, all relevant documents have been attached to make the paper robust. However, the comment regarding including old papers, that are thirty years and above could not be addressed because it will lead to conducting the study from scratch. This should be included as a limitation of the study and a recommendation that further research should conduct a ten-year review of literatures. Results from recent searches will compare if the same results and conclusions can be drawn.

7. PLOS authors have the option to publish the peer review history of their article (what does this mean?). If published, this will include your full peer review and any attached files.

**Do you want your identity to be public for this peer review?** For information about this choice, including consent withdrawal, please see our Privacy Policy.

Reviewer #2: **Yes: **Andrew-Bassey, Uduak Ima
